# Suitability of Chinese oak silkworm eggs for the multigenerational rearing of the parasitoid *Trichogramma leucaniae*

Wei Xu[1], Xuan-Ye Wen[1,2], Yang-Yang Hou[1], Nicolas Desneux[3], Asad Ali[4], Lian-Sheng Zang [1]*

1 Institute of Biological Control, College of Plant Protection, Jilin Agricultural University, Changchun, China,
2 General Station of Forest and Grassland Pest Management, National Forestry and Grassland Administration, Shenyang, China, 3 INRAE, CNRS, UMR ISA, Université Côte d'Azur, Nice, France,
4 Department of Agriculture, Abdul Wali Khan University, Mardan, Pakistan

* lsz0415@163.com

**Data Availability Statement:** The data necessary to replicate the results alongside the manuscript see S1 Dataset in Supporting information.

**Funding:** LSZ was partially supported by the National Key R&D Program of China

## Abstract

*Trichogramma leucaniae* is believed to be an efficient biological control agent for controlling the soybean pod borer [SPB; *Leguminivora glycinivorella*]. The large eggs of Chinese oak silkworm, *Antheraea pernyi*, are one of the best alternative host for mass production of *Trichogramma*. However, they are considered poor host for the growth and development of *T. leucaniae*. Here, we determine the feasibility of successive rearings of *T. leucaniae* on the large eggs for eight generations and evaluated their capacity of parasitizing SPB eggs of different ages. In the first four generations, the suitability of *T. leucaniae* reared on large eggs exhibited a significant increasing tendency and then decreased with the successive generations thereafter. The percentage of parasitized eggs and number of emerged adults per egg were increased from 40.0% and 10.8 adults/egg in F1 generation to 86.7% and 36.4 adults/egg in F4 generation respectively. In addition, *T. leucaniae* reared on *A. pernyi* for four generations significantly parasitized more SPB eggs regardless of egg age compared with those reared on *Corcyra cephalonica* eggs. These results provided useful information on the feasibility of mass production of *T. leucaniae* by reared for successive generations on *A. pernyi* large eggs.

## Introduction

The soybean pod borer [SPB; *Leguminivora glycinivorella* (Matsumura) (Lepidoptera: Tortricidae)] is one of the most serious pests associated with soybean in Northeastern Asia including North of China, Japan, Korea and Russia [1,2]. The SPB lifecycle and soybean pod and seed developmental stages usually occur at the same time. Generally, the adult SPB oviposit on young bean pods from mid-July till the end of August. Upon hatching the young larvae enter the pods and start feeding therein using the immature seeds as a food source until they develop into mature larvae [1]. Soybean pod borer could damage 10%-20% of soybean seeds annually [3]. However, the damage could be over 30–40% if a pest outbreak occurs and timely

(2017YFD0201000). YYH was supported by the National Natural Science Foundation of China (31901946). WX was supported by the Department of Science and Technology, Jilin Province, China (20180201015NY and 20190103102JH). The funders had no role in study design, data collection and analysis, decision to publish, or preparation of the manuscript.

**Competing interests:** The authors have declared that no competing interests exist.

preventive measures are not implemented properly [1]. In addition, SPB infestations also deformed the seeds and thus affect quality and sale prices of the soybean harvest.

In the last three decades, various chemical insecticides have been extensively used to control SPB in soybeans plantations [4]. The exposed period of larvae before they enter the pods (roughly about a single day) is not enough for contact insecticides to work out properly [1], thus it is difficult for the insecticides to contact and kill the pest under a closed canopy [4,5]. In addition, insecticides adversely affect the beneficial natural enemies [6–9], induce the development of insecticide-resistant populations of pest insects and cause environmental pollution [10–12]. Therefore, it is pivotal to look for measures that control the pest and prevent the crop economic loss and also have no or less adverse effect on natural enemies, environment and human health. In this respect, modifying genes using plant-mediated RNA-interference is a promising strategy for controlling SPB infestations [1]. However, the somewhat conservative polices and legal obstacles to transgenic crops in many countries limited the applications of this procedure in soybean [13]. Similarly, certain sex pheromones have also been chemically identified and used to monitor population dynamics and control SPB by mating disruption [14]. The use of resistant cultivars is also an effective way for reducing insect pest damage in combination with other non-chemical control measures [15]. Unfortunately these pest control methods cannot fully suppress soybean pod borer and there is still need of some more effective and sustainable measures to keep the pest under check for long time to get substantial yield of soybean.

The genus *Trichogramma* is endoparasitoid of insect eggs and there are approximately 650 species of egg parasitoids worldwide, of which more than 200 species can parasitize the eggs of crop and forest pests [16]. Successful control in the field has been reported in several countries using *Trichogramma* against the European corn borer, *Ostrinia nubilalis* (Hübner) [17], the polyphagous grapevine moth, *Lobesia botrana* [18], cabbage pests [19,20], *Tuta absoluta* on tomato [21–23], and the oriental fruit moth *Grapholita molesta* [24].

In China, a mass production system is being developed for *Trichogramma* spp. to maintain good quality parasitoids to augment the successful biological control applications for various crops [25]. Several wasp's species are mass-produced and released in large quantities e.g. *Trichogramma dendrolimi* to control the Asian corn borer *Ostrinia furnacalis* [26,27]. In the 1980s and 1990s, *T. dendrolimi* is inundatively released annually across 0.2 to 0.35 million ha of corn to suppress *O. furnacalis* and was increased to 2.3 million ha annually since 2012 in Jilin, China [27]. To attempt to use *Trichogramma* against SPB, indigenous SPB egg parasitoids were initiated in late 2010. Three *Trichogramma* species (*Trichogramma chilonis* Ishii, *Trichogramma ostriniae* Pang & Chen, and *Trichogramma leucaniae* Pang & Chen) were collected from northeastern China soybean fields [28,29]. Through testing their suitability on eggs of different-age SPB, eventually *T. leucaniae* was found to be the most valuable biological control agent against the Soybean pod borers [28,30].

*Trichogramma* parasitoids are mostly produced on factitious hosts, usually on the small eggs of *Corcyra cephalonica*, *Ephestia kuehniella* and *Sitotroga cerealella* which are not economical [31,32]. Small egg hosts, especially *C. cephalonica*, can only produce a wasp per egg, which leads to higher production costs. Moreover, storage of these eggs for more than 2–3 weeks at low temperatures can decrease their suitability for mass rearing [33]. The large eggs of *Antheraea pernyi* are considered to be better factitious hosts with some practical advantages, such as a higher *Trichogramma* reproduction rates with 60–70 wasps per egg [33]. Larger *A. pernyi* eggs also offer increased convenience for storage and transportation and thus can lower the production costs, compared to the small eggs used for the mass rearing of *Trichogramma* parasitoids [33]. However, some limiting factors e.g. thick and hard chorion of *A. pernyi* eggs

make it difficult for some parasitoids, such as *T. ostriniae*, *T. evanesens*, and *T. embryophagum*, adults to make a hole and crawl out of the eggs upon completion of their life cycle [25,34]. For other parasitoids such as *T. leucaniae*, only a low percentage of *A. pernyi* eggs were parasitized that resulted in low parasitoid emergence [35]. Generally, most of the *Trichogramma* parasitoids cannot be massively produced using the eggs of *A. pernyi* except for *T. dendrolimi* and *T. chilonis* [36].

However, the laboratory studies showed that *Trichogrammatoidea hypsipylae* was the most suitable candidate to control *Conopomorpha sinensis*, but it could not emerge from the parasitized eggs of *C. sinensis*. After *T. hypsipylae* were successively reared for 35 generations on the eggs of *C. cephalonica*, however, the wasps could successfully emerge from *C. sinensis* eggs [37]. Therefore, in the present study, we attempted to domesticate the breeding of *T. leucaniae* on large eggs of *A. pernyi* by successive generations and compare their parasitism and suitability on different-aged SPB eggs with those reared on the small eggs of *C. cephalonica*. The objective was to provide valuable techniques and information for mass production of *T. leucaniae* to control SPB.

## Materials and methods

### Parasitoids

The parasitoids, *Trichogramma leucaniae* were originally collected from soybean fields from sentinel SPB eggs in Heihe, Heilongjiang Province, China. The collection of *T. leucaniae* did not involve endangered or protected species, and no specific permissions were required. The parasitoid species was identified using SEM micrographs by examining male genital capsules [38], and then confirmed the analysis by rDNA-ITS2 sequences (GenBank Accession no. HG518480) [39]. The collected specimens were then deposited in the collection of the Institute of Biological Control, Jilin Agricultural University, Changchun, China. The parasitoid colony has been maintained since 2011 using rice moth *Corcyra cephalonica* eggs under laboratory conditions at 26 ± 1 °C and 70±5% RH, with a 16:8 L:D photoperiod. Meanwhile, the *T. leucaniae* population was revitalized every year in August by allowing them to parasitize native SPB host eggs.

### Factitious hosts

Chinese oak silkworm: *Antheraea pernyi* cocoons were collected in Jilin, China and stored at -4 °C for 2 to 3 months. In spring 2018, the cocoons were hanged in an emergence room for incubation (about 25 d) at 25 °C. After adult emergence from the cocoons, the female silkworm moths were collected. Then, eggs of the experiment were collected by dissecting the abdomen of a mature female moth and were washed with distilled water. Immature green eggs were removed, and healthy eggs were then air dried under laboratory conditions. The fresh eggs were used within 8 h of collection for the mass production of *T. leucaniae* and subsequent experimentation.

Rice moth: To obtain host eggs of *Corcyra cephalonica* for breeding of *T. leucaniae*, larvae were placed in a plastic container where they were fed with a mixture of corn flour and wheat bran [40]. Adult moths were collected after emergence and were transferred to an aluminum gauze cage for oviposition. The aluminum gauze cage was kept in an egg collection tray. The moths laid their eggs on the cage walls, which were brushed into the tray at regular intervals with a collecting brush. Collected rice moth eggs were filtered through a screening net of approximately 0.5 mm to remove scales and other debris of the rice moth to obtain clean eggs. The rice moth eggs were collected daily and exposed to 30w UV light for 40 minutes to kill the embryos for further use in breeding of *T. leucaniae*.

## Experimental host

Soybean pod borer: The SPB adults were collected using an insect-collecting net from a soybean field in Jilin Agricultural University, Changchun, China during August 2018. Twenty SPB moths were randomly selected and placed in separate clear plastic cups (10 cm in diameter, 15 cm in height). Furthermore, they were provided with a water-cultured top soybean pod, which included five pods. After 12 h, the adults were removed, and 0-, 2- and 4-day-old eggs of SPB on the soybean pods were selected as test hosts according to the procedure described by Song et al. [28] for obtaining various ages of host eggs.

## Rearing of *T. leucaniae* on *A. pernyi* eggs by successive generations

The experiment was conducted under laboratory conditions at 26±1 ˚C, 70±5% RH and photoperiod of 16:8 L:D. One newly emerged female of *T. leucaniae* from a small rice moth egg, with no prior exposure to the larger *A. pernyi* eggs and mated within 8 h, was introduced into a glass tube (3.5 cm in diameter and 10 cm long) with one large egg of *A. pernyi*. After 24 h, the parasitoid adults were removed and the parasitized eggs of *A. pernyi* were collected. The parasitized eggs were then put in an incubator under the same environmental conditions to allow the next generation of parasitoids to develop. Parasitized eggs were monitored daily until all adult parasitoid emergence had ceased. The numbers of emerged female and male parasitoids per large egg were recorded. The pre-emergence time of each emerged parasitoid was also calculated and recorded as the number of days from parasitism to adult parasitoid emergence from the host. When no parasitoids emerged, all test host eggs were dissected, and recorded the number of unemerged adults for each egg. The *T. leucaniae* progeny that initially emerged from the large eggs of *A. pernyi* were recorded as the first generation (F1) after the mother *T. leucaniae* reared on the rice moth eggs parasitized *A. pernyi* eggs. The newly emerged F1 (until F7) females within 8 h of their emergence were individually introduced into a glass tube with one large *A. pernyi* egg. All biological parameters, such as % of parasitism and emergence, pre-emergence time, number of emerged and unemerged adults, and % of female progeny from F2 to F8 were investigated as per the procedure followed for F1 generation. The tests were replicated 30 times for each generation.

## Parasitism of *T. leucaniae* reared on eggs of *A. pernyi* and *C. cephalonica* on different-aged SPB eggs

The experiment was conducted in an insectary under the same environmental conditions mentioned above. Based on the results of "*T. leucaniae* reared on *A. pernyi* eggs" for 8 successive generations above, *T. leucaniae* of the F4 generation exhibited the best suitability on the large eggs of *A. pernyi*. In order to determine their efficacy on SPB, we compared the parasitism capacity of F4 female parasitoids reared on *A. pernyi* eggs on different aged eggs of SPB with *T. leucaniae* reared on *C. cephalonica*. Individual adult females reared on eggs of *A. pernyi* or *C. cephalonica* were introduced into separate clear plastic cups with 60–100 SPB eggs on soybean pods. Each cup contained 0-, 2-, or 4-d-old SPB host eggs. After 24 h of exposure, the female parasitoid was removed. The experiment was replicated 20 times for *T. leucaniae* reared on large egg and small eggs at different egg ages, respectively. The number of parasitized SPB eggs from each cup was counted under a stereoscopic microscope 6 days after the parasitoid was removed.

## Data analyses

A one-way analysis of variance (ANOVA) was conducted to determine the significance of the differences in the pre-emergence time, number of emerged and unemerged adults, and

percentage of female progeny of *T. leucaniae* reared on large eggs for eight generations. The impacts of the rearing host species and the host age on the number of host eggs parasitized were analyzed by a two-way ANOVA. All data were subjected to normality testing (Shapiro–Wilk test) before performing their ANOVA. Percentage data were arcsine square roots transformed, and count data were logarithm-transformed prior to the normality tests. The means of each parameter were separated using Tukey's honestly significant difference (HSD) test at P < 0.05. All data were statistically analyzed using Data Processing System (DPS, v13.5) software.

## Results

### Parasitism and suitability of *T. leucaniae* reared on *A. pernyi* eggs for successive generations

*Trichogramma leucaniae* from small eggs of *C. cephalonica* were successively reared on large eggs of *A. pernyi* for 8 generations. There were significant differences in parasitism ($F_{7, 232} = 8.47$, $P < 0.0001$) and emergence ($F_{7, 129} = 7.04$, $P < 0.0001$) between the different generations. The percentage of parasitism and percent emergence of *T. leucaniae* from F1 to F4 exhibited a significant increasing tendency (Fig 1). The percentages of parasitized eggs and emergence increased from 40.0% and 38.1% for F1 to 86.7% and 86.3% for F4, respectively. From F5 onwards, the percentages of parasitism and emergence decreased dramatically till F8, where the % of parasitism decreased to 13.3%, and no *T. leucaniae* wasps were observed emerging from the large eggs.

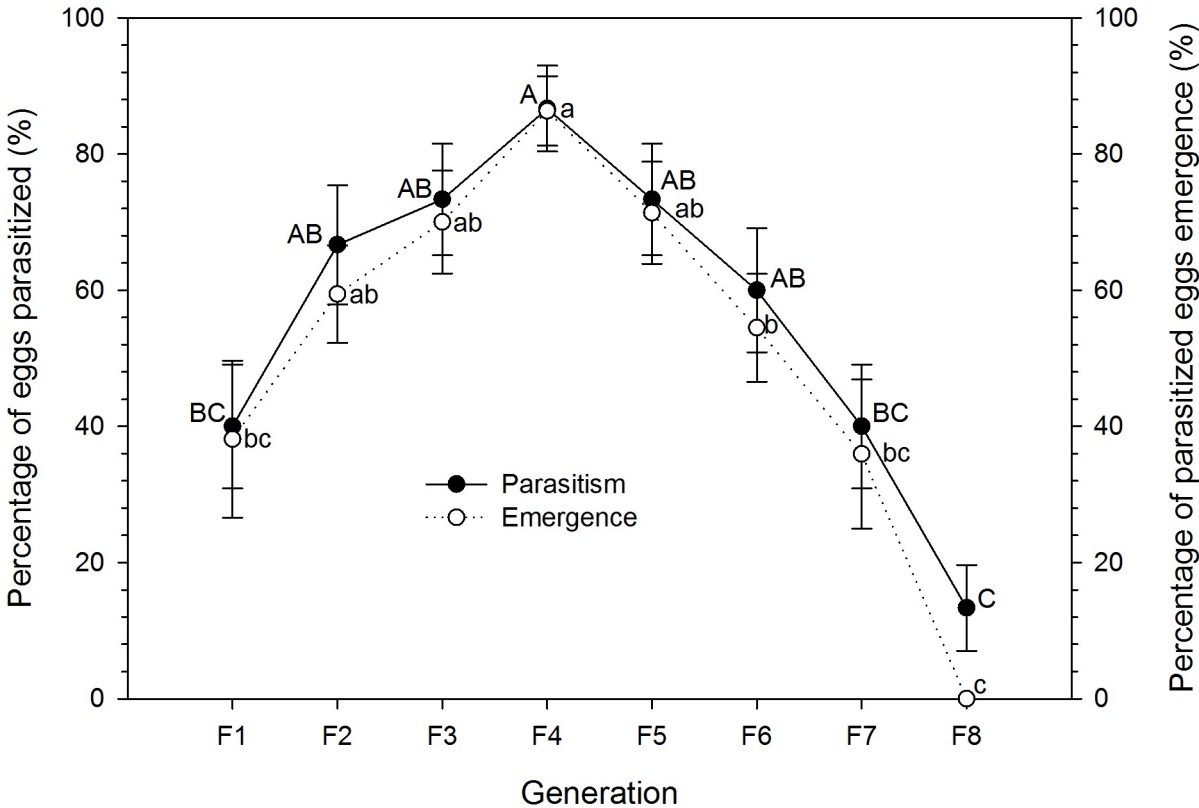

**Fig 1. The percentage of parasitized eggs and the percent emergence of *T. leucaniae* successively reared on large eggs of *A. pernyi* for 8 generations.** Mean±SE are presented. Different upper-case letters on the solid line indicate significant differences in mean percentage of parasitism among different generations. While different lower-case letters on the dotted line indicate significant differences in mean percentage of emergence among different generations. Means were separated with Tukey HSD test at P < 0.05.

The pre-emergence times of *T. leucaniae* reared on the large eggs of *A. pernyi* were significantly different between different generations (Table 1). The pre-emergence time gradually shortened in the first five generations and began to prolong from F6 generation onward. Generally, F1 had the longest pre-emergence time (14.11 d), followed by F2 and F7, and F3-F6 had the shortest pre-emergence time (12.95–13.08 d) respectively, while no parasitoid could fully emerge during the F8 generation.

The number of emerged adults per egg increased with the generation number among the first four generations and then decreased gradually from F5 to F7, and became equal to zero at F8. There was a significant difference in the number of emerged adults per egg among the different generations (Table 1). The number of emerged adults per egg increased from 10.8 during F1 to 36.4 during F4, with the latter value being significantly more important than values recorded during the other generations (F1, F2, F3, F5, F6, F7 and F8).

There was also a significant difference in female progeny among the different generations (Table 1). A clear tendency was shown about the percentage of female progeny that gradually increased in the first five generations and then decreased. F5 had significantly the largest percentage of female progeny (80.76%) but was followed by F4 with (79.11%) of female progeny, in comparison to all the other generations.

The number of unemerged adults per egg decreased with increasing generation numbers among the first four generations and then increased from F5 to F7. Likewise, there were significant differences in the number of unemerged adults among different generations (Table 1). The largest numbers of unemerged adults per egg (19.33) were observed in F1 and F7, which were significantly greater than those of F2, F3 and F4. However, the number of unemerged adults per egg for F4 was the lowest (4.46), significantly which was less than those of any other generation.

## Parasitism of *T. leucaniae* reared on eggs of *A. pernyi* and *C. cephalonica* on different-age eggs of SPB

There were significant impacts of both the rearing host species ($F_{1, 114} = 24.02$, $P < 0.0001$) and age ($F_{2, 114} = 17.49$, $P < 0.0001$) on the number of host eggs parasitized by *T. leucaniae*. However, their interactions did not show any significant influence on the number of host eggs parasitized by *T. leucaniae* ($F_{2, 114} = 1.48$, $P = 0.2310$). *T. leucaniae* reared on *A. pernyi* parasitized significantly more SPB eggs at all egg ages compared to those reared on *C. cephalonica* (Fig 2). There were respectively 39.6, 30.5 and 21.5 of fresh, 2 and 4 day old SPB eggs parasitized by *T. leucaniae* reared on A. pernyi compared to 24.4, 23.8 and 12.7 of fresh, 2 and 4 day

**Table 1. Comparisons of pre-emergence time, number of emerged adults, female progeny and number of adults unemerged of *T. leucaniae* reared on large eggs of *A. pernyi* by successive generations.**

| Generation | Parameters | | | |
|---|---|---|---|---|
| | Pre-emergence time (days) | No. of emerged adult/egg | Female progeny (%) | No. of adults unemerged |
| F1 | 14.11±0.13 a | 10.83±3.40 c | 63.73±2.37 e | 19.33±4.15 a |
| F2 | 13.55±0.28 b | 22.00±2.87 bc | 69.39±1.92 de | 12.70±1.80 ab |
| F3 | 13.08±0.10 c | 29.00±3.33 ab | 75.57±1.49 bc | 9.82±2.17 b |
| F4 | 13.08±0.11 c | 36.38±2.72 a | 79.11±1.44 ab | 4.46±1.18 c |
| F5 | 12.95±0.07 c | 28.91±3.46 ab | 80.76±1.23 a | 8.82±1.95 bc |
| F6 | 12.96±0.05 c | 20.33±3.29 bc | 71.64±3.06 cd | 13.33±1.69 ab |
| F7 | 13.09±0.13 bc | 12.00±3.99 c | 66.07±4.01 de | 19.33±3.30 a |
| | $F_{6, 101} = 4.70$ P = 0.0003 | $F_{6, 131} = 7.39$ P < 0.0001 | $F_{6, 101} = 8.27$ P < 0.0001 | $F_{6, 131} = 6.00$ P < 0.0001 |

For each parameter, Means ± SE are shown. Means in a column followed by different letters indicate significant differences (Tukey's test, P < 0.05).

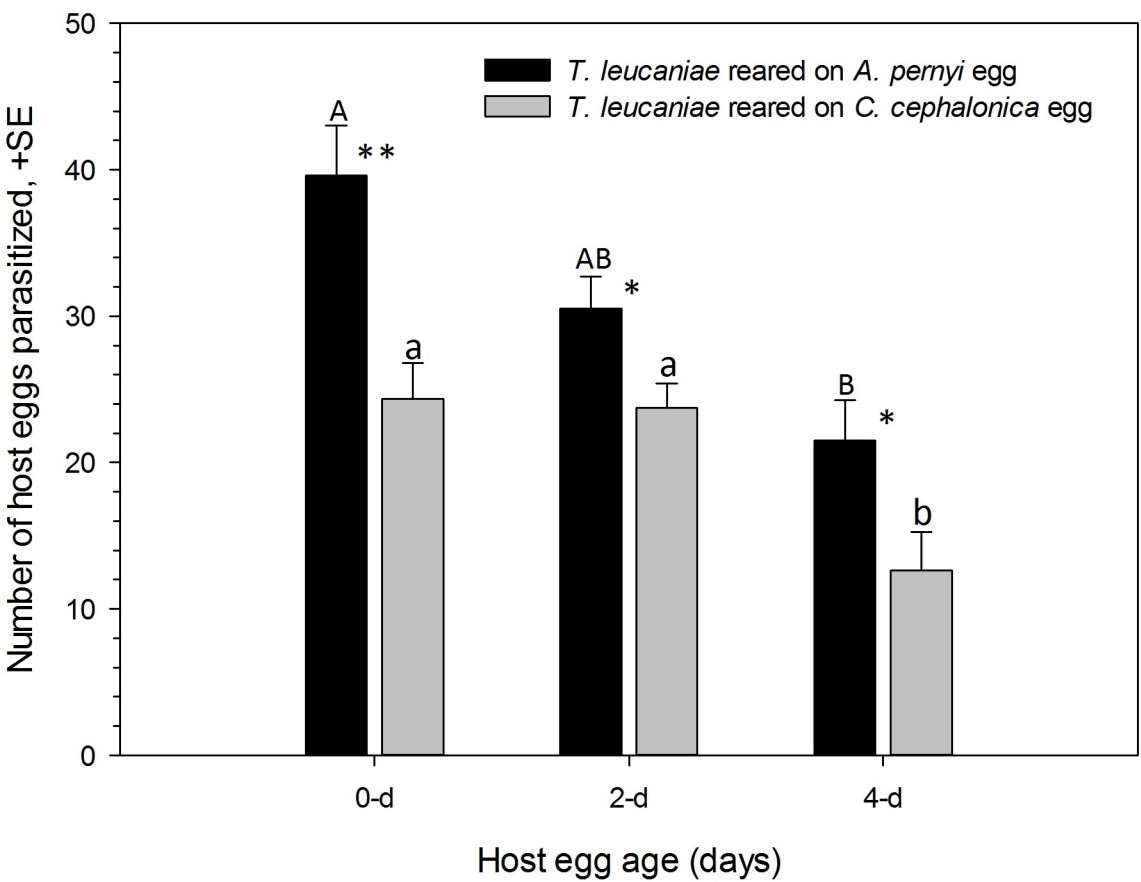

**Fig 2. Comparisons of different aged eggs of *L. glycinivorella* parasitized by *T. leucaniae* reared on eggs of *A. pernyi* and *C. cephalonica*.** Different uppercase letters on the black bars indicate significant differences in eggs mean (± SE) of different ages parasitized by *Trichogramma* reared on *A. pernyi* eggs. Different lowercase letters on the gray bars indicate significant differences in eggs mean (± SE) of different ages parasitized by *Trichogramma* reared on *C. cephalonica* eggs. The paired bars with an asterisk and two asterisks indicate significant differences between the means (± SE) at $P < 0.05$ and $P < 0.001$, respectively.

old SPB eggs parasitized by *T. leucaniae* reared on *C. cephalonica* eggs, respectively. In addition, *T. leucaniae* reared on *A. pernyi* mostly parasitized 0-day-old SPB eggs followed by 2-day-old with the lowest parasitism being recorded on 4-day-old eggs. However, *T. leucaniae* reared on *C. cephalonica* parasitized similar numbers on 0- and 2-day-old eggs, being both significantly more than the number obtained on 4-day-old eggs.

## Discussion

*Trichogramma leucaniae* could be reared on the large eggs of *A. pernyi* with divergent adaptation for 8 successive generations, showing that the biological traits of parasitoids can be changed through artificial domestication. A previous study confirmed that parasitoid host acceptance behavior is flexible and shows a change from the inadaptation to the adaptation on the new host and new environment [41]. Moreover, the host adaptation is different among various species of parasitoids. For example, the rearing of *Trichogramma semifumatum* on *Sitotroga cerealella* eggs for 3 generations weakened their host preference to *Trichoplusia ni* and *S. cerealella* eggs [42], whereas some other *Trichogramma* species need rearing for many

generations to change their preferences and other traits to adapt to the new hosts and exhibit high parasitic capacities [43].

In our study, however, the unsuitability of *T. leucaniae* was found to increase with increasing generations after being successively reared on *A. pernyi* for 4 generations. This was noticed by the parasitism rate that decreased, and the pre-emergence time that was prolonged. The same phenomenon also existed in the continuous rearing of *Trichogramma* on other alternative hosts. For instance, after rearing on *Anagasta kuehniella* (Zeller) for 32 generations, the parasitism rate of *T. pretiosum* decreased and the developmental time was prolonged [43]. Host unsuitability is also exhibited in the agility of parasitoids. In fact, Dutton and Begler [44] reported that rearing *Trichogramma brassicae* Bezdenko on *Ephestia kuehniella* Zeller eggs for more than 39 generations resulted in deformed wings and poor flying capacity of the parasitoid wasps. Similarly, a decline in quality has been observed in *Trichogramma* reared continuously in the laboratory on the same factitious host [45]. This shows that the adaptability of parasitoids decreases, and their populations degenerate on alternative hosts after reproduction for many generations, due to various factors such as host's nutritional quality. Therefore, it is necessary to periodically introduce field individuals to reduce the possible losses of parasitoid quality during mass production and augmentative biological control of pests using *Trichogramma* [43].

On the other hand, the large eggs of *A. pernyi* can produce more wasps, with the most of them being able to mate before emerging. Therefore, inbreeding is very common in *Trichogramma* reared on large eggs, which may cause *Trichogramma* depression and thus increases their unsuitability. Of course, several studies have documented that *Trichogramma* are not subjected to inbreeding depression [46,47]. However, Antolin [48] in his study showed that the total fecundity of inbred *T. pretiosum* females represented only 78% of outcrossed females fecundity, thus indicating inbreeding depression. According to Sorati et al. [49], an inbred strain still adapts much more slowly to an artificial environment than an outbred strain because of the initial low levels of genetic variance for traits under selection. Hence, more work is needed to confirm the effects of inbreeding on *T. leucaniae*.

According to Harvey [50], inconstant states and variable conditions of host quality, such as species, size, age and nutritional status, can affect the parasitoid growth, development and survival. So, as the *Trichogramma* wasps generally rely on a limited amount of resources to complete their development, selecting an optimum host plays a pivotal role in their reproduction. Our results showed that the numbers of SPB eggs of different ages parasitized by the fourth generation of *T. leucaniae* reared on *A. pernyi* were significantly higher than those parasitized by the wasps reared on small eggs of *C. cephalonica*, which is consistent with previous studies. For instance, Liu et al. [51] reported that the parasitization capacity of *T. dendrolimi* females reared on *A. pernyi* was significantly higher than those reared on *C. cephalonica* eggs, as shown by the differences in the percentage of females with successful parasitization, the number of host eggs parasitized per female, and the percentage of host eggs parasitized in 24 h. Moreover, *Trichogramma* emerging from large hosts were larger in size and developed faster than those emerging from small hosts [51,52]. This is due to the fact that large hosts provide more nutrition and thus produce larger and stronger offspring, which might be able to fly farther per unit time and might be more efficient in searching and attacking hosts in the field than smaller individuals. Hence resultantly, larger wasps are more fecund, long-lived, parasitize more hosts, and produce more progeny than their smaller conspecifics [53,54].

Host age is an important factor that influences the availability and parasitoid acceptance of hosts. Generally, younger eggs have more yolk and are more nutritious, and are preferred by most *Trichogramma spp.* [28,55]. Hou et al. [56] reported five *Trichogramma* species that

parasitized *Mythimna separata* at all egg ages, but also reported that three *Trichogramma* species, namely *T. ostriniae*, *T. japonicum*, and *T. leucaniae* showed higher parasitism efficiency only on 0-day-old eggs. In the present study, *T. leucaniae* reared on *A. pernyi* also parasitized the most SPB eggs at the age of 0-day-old. The number of host eggs parasitized decreased with the increasing age of the host eggs. Although there were similar numbers of fresh and 2 day old host eggs parasitized by wasps reared on *C. cephalonica*, we notice that the rearing host distinctly influenced the wasp parasitic capacity and accurate identification for the age of host eggs. *Trichogramma leucaniae* reared on *A. pernyi* eggs not only parasitized more SPB eggs than the wasps reared on *C. cephalonica* but also had a stronger ability to recognize SPB eggs of different ages, which is beneficial for their offspring quality. This finding indicates that *A. pernyi* egg is a potential alternative host for *T. leucaniae* mass production.

These results further confirmed previous studies that *T. leucaniae* has an increased parasitic capacity and suitability [28,30], and appears to be an effective augmentative biological control agent against SPB. *Trichogramma leucaniae* rearing based on large eggs of *A. pernyi* make it possible to mass-produce and inundatively release them against SPB. In addition, *T. leucaniae* can parasitize other lepidopteran pests that occur and cause damage simultaneously with SPB. Furthermore, *T. leucaniae* also exhibited better parasitic capacity and suitability on *Ascotis selenaria* eggs, one of the most important defoliators of soybean than the other two *Trichogramma* species present in soybean fields [57]. Therefore, releasing *T. leucaniae* is not only effective for SPB control but also *A selenaria* and other pests, which exhibits that *T. leucaniae* is an economic and effective biological control agent. However, to improve the efficacy of control, field application and release techniques based on *T. leucaniae* and SPB biological characteristics should be further studied in detail, as multiple factors could impact effectiveness of *Trichogramma* parasitoids [58–65].

## Supporting information

**S1 Dataset. The data necessary to replicate the results alongside the manuscript.**
(XLSX)

## Author Contributions

**Conceptualization:** Lian-Sheng Zang.

**Data curation:** Wei Xu, Xuan-Ye Wen, Yang-Yang Hou, Lian-Sheng Zang.

**Formal analysis:** Yang-Yang Hou, Lian-Sheng Zang.

**Funding acquisition:** Wei Xu, Lian-Sheng Zang.

**Investigation:** Wei Xu, Xuan-Ye Wen.

**Methodology:** Lian-Sheng Zang.

**Project administration:** Lian-Sheng Zang.

**Resources:** Lian-Sheng Zang.

**Supervision:** Lian-Sheng Zang.

**Writing – original draft:** Wei Xu, Lian-Sheng Zang.

**Writing – review & editing:** Wei Xu, Nicolas Desneux, Asad Ali, Lian-Sheng Zang.

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
