## [Decision Letter · Decision Letter 0]

5 Mar 2020

PONE-D-20-03548

Suitability of Trichogramma leucaniae reared on large eggs of Chinese oak silkworm, Antheraea pernyi for successive generations

PLOS ONE

Dear Dr. Zang,

Thank you for submitting your manuscript to PLOS ONE. After careful consideration, we feel that it has merit but does not fully meet PLOS ONE’s publication criteria as it currently stands. Therefore, we invite you to submit a revised version of the manuscript that addresses the points raised during the review process.

Your manuscript has been reviewed by two independent and qualified reviewers. Both of them found the paper publishable in PlosOne after minor revisions. I concur with them, thus I invite you to follow the comments below and to submit the revised version of the manuscript (track changes and clean versions) along with a point by point rebuttal letter.

Authors declared that data are available witouth restriction, but I was not able to understand how. Can authors read and follow the data availability PlonOnse roles https://journals.plos.org/plosone/s/data-availability ?

Title: I found the title unclear and in any case missing of a comma after the specific name. The word ‘suitability’ as it is used is confusing: is it referred to the parasitoid or to the host? Also, ‘large eggs’ should not be used in the title, because it is misleading: did authors select the bigger eggs from the same host? I do not think so. Did authors test eggs of different size? It seems they do not.

Authors should find a more suitable and clear title. An example could be

“Suitability of Chinese oak silkworm eggs for the multigenerational rearing of the parasitoid Trichogramma leucaniae”

We would appreciate receiving your revised manuscript by Apr 19 2020 11:59PM. To enhance the reproducibility of your results, we recommend that if applicable you deposit your laboratory protocols in protocols.io, where a protocol can be assigned its own identifier (DOI) such that it can be cited independently in the future. For instructions see: http://journals.plos.org/plosone/s/submission-guidelines#loc-laboratory-protocols

We look forward to receiving your revised manuscript.

Kind regards,

Antonio Biondi, Ph.D.

Academic Editor

PLOS ONE

Additional Editor Comments (if provided):

Your manuscript has been reviewed by two independent and qualified reviewers. Both of them found the paper publishable in PlosOne after minor revisions. I concur with them, thus I invite you to follow the comments below and to submit the revised version of the manuscript (track changes and clean versions) along with a point by point rebuttal letter.

Authors declared that data are available witouth restriction, but I was not able to understand how. Can authors read and follow the data availability PlonOnse roles https://journals.plos.org/plosone/s/data-availability ?

Title: I found the title unclear and in any case missing of a comma after the specific name. The word ‘suitability’ as it is used is confusing: is it referred to the parasitoid or to the host? Also, ‘large eggs’ should not be used in the title, because it is misleading: did authors select the bigger eggs from the same host? I do not think so. Did authors test eggs of different size? It seems they do not.

Authors should find a more suitable and clear title. An example could be

“Suitability of Chinese oak silkworm eggs for the multigenerational rearing of the parasitoid Trichogramma leucaniae”

Journal Requirements:

2. In your Methods section, please provide additional location information of the collection sites, including geographic coordinates for the data set if available.

3. In your Methods section, please provide additional information regarding the permits you obtained for the work. Please ensure you have included the full name of the authority that approved the collection sites access and, if no permits were required, a brief statement explaining why.

Reviewers' comments:

Reviewer's Responses to Questions

**Comments to the Author**

1. Is the manuscript technically sound, and do the data support the conclusions?

Reviewer #1: Yes

Reviewer #2: Yes

2. Has the statistical analysis been performed appropriately and rigorously? 

Reviewer #1: Yes

Reviewer #2: Yes

3. Have the authors made all data underlying the findings in their manuscript fully available?

Reviewer #1: Yes

Reviewer #2: Yes

4. Is the manuscript presented in an intelligible fashion and written in standard English?

Reviewer #1: Yes

Reviewer #2: Yes

5. Review Comments to the Author

Reviewer #1: The manuscript “Suitability of Trichogramma leucaniae …” reported the outcome of rearing Trichogramma leucaniae successively on eggs of Antheraea pernyi for several generations, and compared the control efficacy of the parasitoids with those reared on eggs of Corcyra cephalonica in parasitizing eggs of Leguminivora glycinivorella. The data support the conclusions and the authors made all data underlying the findings. The manuscript provides interesting information of the mass production of T. leucaniae on large eggs. I think the manuscript can be accepted for publication after having a minor revision.

The title needs to be reconsidered. It does not reflect the content of the manuscript. For example, suitable to what or for which?

Line 33 ‘pest’ should be ‘pests’;

Line 46 delete the second ‘thus’;

Lines 178 I don’t understand why you provided different amount of eggs to individual parasitoid?

Lines 270-271 Why did you say the reproductive index is high?

Reviewer #2: Line 99: “However, the laboratory studies…” if “However” added in, this sentence may be a more reasonable and logical connection in the context.

Line 159-160: “When no parasitoids… unemerged adults.” Only the host eggs with no emergence of parasitoids were dissected? How about the rate of the other eggs with partly emergent wasps? Do you mean either all parasitoids emerge or all died within an egg?

Line 194: “Data … software” the year or the edition of this soft should be provided.

Line 249-251: “Trichogramma … could be … domestication” Only four generations without other evidences, this statement is a little bit of subjective. Please reconsider the sentence.

Line 259-260: “In our study… for generations” It seems contradictory statement compared to the statement mentioned above in Line 249-251. Please reconsider what did you mean by this.

6. PLOS authors have the option to publish the peer review history of their article (what does this mean?). If published, this will include your full peer review and any attached files.

Reviewer #1: No

Reviewer #2: No

---

## [Author Response · Author response to Decision Letter 0]

11 Mar 2020

Responses on the comments of reviewers

Comments of the editor

Thank you for submitting your manuscript to PLOS ONE. After careful consideration, we feel that it has merit but does not fully meet PLOS ONE’s publication criteria as it currently stands. Therefore, we invite you to submit a revised version of the manuscript that addresses the points raised during the review process.

Your manuscript has been reviewed by two independent and qualified reviewers. Both of them found the paper publishable in PlosOne after minor revisions. I concur with them, thus I invite you to follow the comments below and to submit the revised version of the manuscript (track changes and clean versions) along with a point by point rebuttal letter.

Answer: Thanks Dr. Antonio Biondi for the active comments on our study. We carefully revise the paper according to the reviewers and editors, see the following responses.

Authors declared that data are available without restriction, but I was not able to understand how. Can authors read and follow the data availability PlonOnse roles https://journals.plos.org/plosone/s/data-availability ?

 Answer: I conducted a wrong operation during the submission.

Title: I found the title unclear and in any case missing of a comma after the specific name. The word ‘suitability’ as it is used is confusing: is it referred to the parasitoid or to the host? Also, ‘large eggs’ should not be used in the title, because it is misleading: did authors select the bigger eggs from the same host? I do not think so. Did authors test eggs of different size? It seems they do not.

Authors should find a more suitable and clear title. An example could be

“Suitability of Chinese oak silkworm eggs for the multigenerational rearing of the parasitoid Trichogramma leucaniae”

Answer: Thank you for the valuable comments and revision suggestion. We accept the title that you provided. In this paper, we compared parasitism of T. leucaniae reared with Chinese oak silkworm and rice moth eggs on SPB. In China, it is popular to say large egg for Chinese oak silkworm (diameter >4 mm), small egg for rice moth (diameter <0.6mm). In order to avoid confuse, large egg is not used in the title, but expressed in the text.

Additional Editor Comments (if provided):

Your manuscript has been reviewed by two independent and qualified reviewers. Both of them found the paper publishable in PlosOne after minor revisions. I concur with them, thus I invite you to follow the comments below and to submit the revised version of the manuscript (track changes and clean versions) along with a point by point rebuttal letter.

Answer: Thanks for the active comments on our study. We carefully revise the paper according to the reviewers and editors, see the following responses.

Authors declared that data are available witouth restriction, but I was not able to understand how. Can authors read and follow the data availability PlonOnse roles https://journals.plos.org/plosone/s/data-availability ?

Answer: Sorry for the missing information. For this revision version, we provide S1 Dataset in the Supporting information.

Title: I found the title unclear and in any case missing of a comma after the specific name. The word ‘suitability’ as it is used is confusing: is it referred to the parasitoid or to the host? Also, ‘large eggs’ should not be used in the title, because it is misleading: did authors select the bigger eggs from the same host? I do not think so. Did authors test eggs of different size? It seems they do not.

Authors should find a more suitable and clear title. An example could be

“Suitability of Chinese oak silkworm eggs for the multigenerational rearing of the parasitoid Trichogramma leucaniae”

Answer: Thank you for the valuable comments and revision suggestion. We accept the title that you provided. In this paper, we compared parasitism of T. leucaniae reared with Chinese oak silkworm and rice moth eggs on SPB. In China, it is popular to say large egg for Chinese oak silkworm (diameter >4 mm), small egg for rice moth (diameter <0.6mm). In order to avoid confuse, large egg is not used in the title, but expressed in the text.

Review Comments to the Author

Reviewer #1: The manuscript “Suitability of Trichogramma leucaniae …” reported the outcome of rearing Trichogramma leucaniae successively on eggs of Antheraea pernyi for several generations, and compared the control efficacy of the parasitoids with those reared on eggs of Corcyra cephalonica in parasitizing eggs of Leguminivora glycinivorella. The data support the conclusions and the authors made all data underlying the findings. The manuscript provides interesting information of the mass production of T. leucaniae on large eggs. I think the manuscript can be accepted for publication after having a minor revision.

The title needs to be reconsidered. It does not reflect the content of the manuscript. For example, suitable to what or for which?

Answer: The title is changed to ‘Suitability of Chinese oak silkworm eggs for the multigenerational rearing of the parasitoid Trichogramma leucaniae’

Line 33 ‘pest’ should be ‘pests’;

Answer: Done.

Line 46 delete the second ‘thus’;

Answer: Done.

Lines 178 I don’t understand why you provided different amount of eggs to individual parasitoid?

Answer: During the preparation of natural host eggs of SPB, 20 SPB moths were randomly selected and introduced in clear plastic cup with a water-cultured top soybean pod. After 12 h, a range of 60-100 host eggs deposited on the soybean pod will be obtained. For this experiment, we hardly prepare similar numbers of host eggs to a test parasitoid. More than 40 eggs is superfluous number to Trichogramma in 24, so 60-100 host eggs is enough to be parasitized by a female wasp in 24 h.

Lines 270-271 Why did you say the reproductive index is high?

Answer: Thanks for the valuable comment, it is not suitable to state a high reproductive index on the natural host. The sentence is changed to ‘This shows that the adaptability of parasitoids decreases, and their populations degenerate on alternative hosts after reproduction for many generations, due to various factors such as host’s nutritional quality.’ 

Reviewer #2: Line 99: “However, the laboratory studies…” if “However” added in, this sentence may be a more reasonable and logical connection in the context.

Answer: We have revised the sentence according to the suggestion.

Line 159-160: “When no parasitoids… unemerged adults.” Only the host eggs with no emergence of parasitoids were dissected? How about the rate of the other eggs with partly emergent wasps? Do you mean either all parasitoids emerge or all died within an egg?

Answer: In order to avoid confuse, we change the sentence to ‘When no parasitoids emerged, all test host eggs were dissected, and recorded the number of unemerged adults for each egg.’

Line 194: “Data … software” the year or the edition of this soft should be provided.

Answer: The sentence is changed to ‘All data were statistically analyzed using Data Processing System (DPS, v13.5) software.’

Line 249-251: “Trichogramma … could be … domestication” Only four generations without other evidences, this statement is a little bit of subjective. Please reconsider the sentence.

Answer: Based on the suggestion, we change the sentence to ‘Trichogramma leucaniae could be reared on the large eggs of A. pernyi with divergent adaptation for 8 successive generations, showing that the biological traits of parasitoids can be changed through artificial domestication.’

Line 259-260: “In our study… for generations” It seems contradictory statement compared to the statement mentioned above in Line 249-251. Please reconsider what did you mean by this.

Answer: after the revision on Line 249-251, the statement of Line 259-260 will be rational.

---

## [Editor Report · Decision Letter 1]

17 Mar 2020

Suitability of Chinese oak silkworm  eggs for the multigenerational rearing of the parasitoid Trichogramma leucaniae

PONE-D-20-03548R1

Dear Dr. Zang,

We are pleased to inform you that your manuscript has been judged scientifically suitable for publication and will be formally accepted for publication once it complies with all outstanding technical requirements.

With kind regards,

Antonio Biondi, Ph.D.

Academic Editor

PLOS ONE
---

## [Editor Report · Acceptance letter]

7 Apr 2020

PONE-D-20-03548R1 

Suitability of Chinese oak silkworm  eggs for the multigenerational rearing of the parasitoid Trichogramma leucaniae 

Dear Dr. Zang:

I am pleased to inform you that your manuscript has been deemed suitable for publication in PLOS ONE. Congratulations! Your manuscript is now with our production department. 

With kind regards,

on behalf of

Dr. Antonio Biondi 

Academic Editor

PLOS ONE